# Angle of Arrival for the Beam Detection Method of Spatially Distributed Sensor Array

**DOI:** 10.3390/s25051625

**Published:** 2025-03-06

**Authors:** Shan Zhao, Lei Zhu, Shiyang Shen, Heng Du, Xiangyu Wang, Lei Chen, Xiaodong Wang

**Affiliations:** 1School of Optoelectronic Engineering, Chongqing University of Posts and Telecommunications, Chongqing 400065, China; zhaoshan@cigit.ac.cn (S.Z.); duheng@cigit.ac.cn (H.D.); 2Chongqing Institute of Green and Intelligent Technology, Chinese Academy of Sciences, Chongqing 400714, China; zhulei@cigit.ac.cn (L.Z.); gbs114@mail.ustc.edu.cn (S.S.); wangxiangyu17@mails.ucas.ac.cn (X.W.); chenlei221@mails.ucas.ac.cn (L.C.)

**Keywords:** angle of arrival, spatially distributed sensor array, wide field of view, low-SWaP

## Abstract

Laser space networks are an important development direction for inter-satellite communication. Detecting the angle of arrival (AOA) of multiple satellites in a wide field of view (FOV) is the key to realize inter-satellite laser communication networking. The traditional AOA detection method based on the lens system has a limited FOV. In this paper, we demonstrate a system that uses a spatially distributed sensor array to detect the AOA in a wide FOV. The basic concept is to detect AOA using the signal strength of each sensor at different spatial angles. An AOA detection model was developed, and the relationship of key structural parameters of the spatially distributed sensor array on the FOV and angular resolution was analyzed. Furthermore, a spatially distributed sensor array prototype consisting of 5 InGaAs PIN photodiodes distributed on a 3D-printed structure with an inclination angle of 30° was developed. In order to improve the angle calculation accuracy, a multi-sensor data fusion algorithm is proposed. The experimental results show that the prototype’s maximum FOV is 110°. The root mean square error (RMSE) for azimuth is 0.6° within a 60° FOV, whereas the RMSE for elevation is 0.67°. The RMSE increases to 1.1° for azimuth and 1.7° for elevation when the FOV expands to 110°. The designed spatially distributed sensor array has the advantages of a wide FOV and low size, weight, and power (SWaP), presenting great potential for multi-satellite laser communication applications.

## 1. Introduction

In recent years, low-earth-orbit satellite (LEO) constellations have been developing rapidly. Through inter-satellite laser links, satellites can achieve high-speed data exchange, significantly enhancing global coverage and low-latency communication performance [1,2,3,4]. However, current LEO satellite laser communication terminals mainly operate in a point-to-point configuration, requiring multiple communication terminals for inter-satellite connectivity. This approach significantly increases the satellite’s size, weight, and power (SWaP), as well as system complexity. To address this problem, a laser terminal using a low-switching wide field of view (FOV) transceiver is proposed to achieve one-to-many communication [5,6,7,8]. Through distributed multi-transmitting units, these optical terminals facilitate flexible transmission across multiple beams and angles. Furthermore, the laser communication terminal must possess wide field of view detection capabilities to obtain the approximate position information of target satellites for the requirements of multi-target positioning and tracking, which is used for the multi-beam coarse alignment. With the integrated precise tracking module, the terminal can achieve rapid switching between one-to-many links, ensuring efficient and reliable inter-satellite communication.

The popular approach for angle of arrival (AOA) wide field of view detection involves utilizing a lens system to focus the incident light onto a sensor array or CCD. The position of the beam is calculated by the change of the centroid of the light spot. In 2017, Williams AJ et al. used a 4 × 4 silicon photomultiplier (SiPM) array with a 20° FOV [9]. In 2022, Zhang et al. proposed a detection system that consisted of a fisheye lens and a CCD with a resolution of 640 pixels × 512 pixels. The system’s measurement FOV was 120° × 96°, and the angle error was 0.5° [10]. The lens-based AOA detection has limitations such as a restricted field of view, optical distortions, and large SWaP consumption, which decrease its applicability in multi-satellite laser communication applications [11,12].

Another approach is to detect the AOA using the signal strength of the sensor array at different spatial angles. It has no lens system and consists of multiple photodiode sensors, which ha the advantages of a wide field of view and low-SWaP. In 2017, Eren F et al. designed a hemispherical 5 × 5 photodetector array and evaluated it using Monte Carlo simulations, showing an estimation error within 5° for elevation and azimuth [13]. In 2018, Jakubaszek M et al. proposed a system that used five sensors distributed on the side panels of a truncated pyramid to detect the AOA. Within a 120° field of view, the estimated errors of elevation and azimuth angles were mostly less than 6° [14]. In 2020, Velazco JE et al. proposed an omnidirectional optical communication/navigation device based on a truncated icosahedron with a measurement field of view of 4π and an RMS of 0.076° [15,16]. In 2024, Patel VH et al. designed a deployable optical receiving aperture consisting of five sensors and tested it for six angles. The maximum angle error exceeded 5°, and no two-dimensional test was performed [17,18].

These research institutions have clarified the AOA detection principle of spatially distributed sensor arrays, developed prototypes, and conducted verification experiments, but the impact and optimization design of key parameters of such arrays, data processing methods, and error analysis are not in-depth enough. There is also a lack of systematic design methods for spatially distributed sensor arrays.

In this paper, we first present the principle of AOA detection using the spatially distributed sensor array. Subsequently, a detailed analysis is conducted to evaluate the influence of the structure on the field of view and angular resolution, leading to the optimization of the spatially distributed sensor array design. A multi-sensor data fusion algorithm is proposed to enhance measurement accuracy. Finally, the primary sources of measurement errors in the spatially distributed sensor array were discussed, and error calibration was implemented. The two-dimensional angle of arrival was also tested, and the test results verified the effectiveness of the proposed multi-sensor data fusion algorithm.

## 2. Methods and Design

### 2.1. Detection Principle

The spatially distributed sensor array is a polyhedral structure, with each face equipped with a sensor that has identical spectral responsivity and photosensitive area. The output of the optical sensor has an angular dependence on the incident light [14], as shown in Figure 1a. The effective area of the detector can be expressed as:(1)$$A_{eff}=A\cos\theta$$
where $A_{eff}$ is the effective area of the sensor, A is the area of the sensor, and θ is the angle between the incident light and the sensor normal vector.

The incident light generates photocurrent proportional to the irradiance and the effective area. It can be expressed by the following equation:(2)$$U=PR_{\lambda }R_{L}A_{eff}$$
where U is the voltage signal on the load resistor $R_{L}$, $R_{\lambda }$ is the responsivity of the photodetector at a wavelength of λ, and P is the light flux received by the sensor from normal incident light. AOA is calculated by combining the optical power received by each sensor at different positions in space. The position of each sensor in the spatially distributed sensor array is shown in Figure 1b. The matrix K, composed of the coordinate matrices of the sensor normal vectors, can be expressed as:(3)$$K=\left[n_{1},n_{2},\ldots ,n_{i}\ldots ,n_{j}\right]$$
where $n_{i}$ represents the position coordinate vector of each sensor, which is expressed as:(4)$$n_{i}={\left(x_{i},y_{i},z_{i}\right)}^{T}$$

The optical power $P_{i}$ incident on each sensor is given by:(5)$$P_{i}=\int S\cdot n_{i}da$$
where S is the energy flux density vector of the incident light, which can be expressed in the Cartesian coordinate system as follows:(6)$$S=S\cdot s=S\cdot \left(X,Y,Z\right)=S\cdot \left(\cos\beta \cos\gamma ,\sin\beta \cos\gamma ,\sin\gamma \right)$$
where S is the total energy flux density of the incident light, β is the azimuth angle, and γ is the elevation angle. The output matrix of the spatially distributed sensor array is expressed as:(7)$$P={\left[\begin{array}{ccc} P_{1} & \cdots P_{i}\cdots & P_{j} \end{array}\right]}^{T}=SK^{T}s$$

According to Equation (6), there are three parameters to be measured: S, β, and γ. Therefore, at least three sensor output powers with uncorrelated normal vectors should be selected from K to calculate the AOA. Matrix B is extracted as a block from matrix K. The rank of matrix B satisfies r(B)≥3 and its corresponding power output matrix is $P_{B}$. The vector s can be calculated by:(8)$$s={\frac{1}{S}\left({BB}^{T}\right)}^{-1}BP_{B}=\left(X,Y,Z\right)$$

Thus the β and γ for the angle of arrival are given by: [18]:(9)$$\begin{array}{c} \beta =\tan^{-1}\frac{Y}{X}\\ \gamma =\tan^{-1}\frac{Z}{\sqrt{X^{2}+Y^{2}}} \end{array}$$

### 2.2. Structural Design of the Spatially Distributed Sensor Array

The number and location of sensors have a direct relationship with the field of view and measurement accuracy of the spatially distributed sensor array. We developed a support structure consisting of one top panel and M side panels, where the angle between the top panel and the side panels is α, and each panel is equipped with the same sensor. According to Equations (7) and (8), the rank of matrix K should satisfy:(10)$$r\left(K\right)\geq 3$$

It indicates that the minimum number of M is 2. The side panels are evenly distributed around the top panel, which is beneficial for analyzing the sensor field of view and error. The field of view of a single sensor is denoted as ϕ. In practical applications, the sensor is encapsulated, resulting in ϕ<π. The number of side panels M should satisfy:(11)$$M>\frac{2\pi }{\phi }\;\left(\phi <\pi \right)$$

The commercially packaged photodiodes have a field of view of 120°, so we choose M=4 to design the spatially distributed sensor array and improve the redundancy of the sensors. Figure 2a shows a schematic diagram of a spatially distributed sensor array when M=4. At this time, the matrix $K^{T}$ can be expressed as:(12)$$K^{T}=\left[\begin{array}{c} n_{1}\\ n_{2}\\ n_{3}\\ n_{4}\\ n_{5} \end{array}\right]=\left[\begin{array}{ccc} 1 & 0 & 0\\ \cos\alpha & \sin\alpha & 0\\ \cos\alpha & 0 & \sin\alpha \\ \cos\alpha & -\sin\alpha & 0\\ \cos\alpha & 0 & -\sin\alpha \end{array}\right]$$

Substituting $K^{T}$ into Equation (7), the output power of each sensor can be expressed as:(13)$$P_{1}=P_{1}\left(S,\beta ,\gamma \right)=S\cos\beta \cos\gamma$$(14)$$P_{2}=P_{2}\left(S,\beta ,\gamma \right)=S\left(\sin\alpha \sin\beta \cos\gamma +\cos\alpha \cos\beta \cos\gamma \right)$$(15)$$P_{3}=P_{3}\left(S,\beta ,\gamma \right)=S(\sin\alpha \sin\gamma +\cos\alpha \cos\beta \cos\gamma )$$(16)$$P_{4}=P_{4}\left(S,\beta ,\gamma \right)=S\left(-\sin\alpha \sin\beta \cos\gamma +\cos\alpha \cos\beta \cos\gamma \right)$$(17)$$P_{5}=P_{5}\left(S,\beta ,\gamma \right)=S(-\sin\alpha \sin\gamma +\cos\alpha \cos\beta \cos\gamma )$$

The output power in the field of view of each sensor should be satisfied:(18)$$P_{i}=P\left(S,\beta ,\gamma ,\alpha \right)=S\left(s\cdot n_{i}\right)\geq S\cos\frac{\phi }{2}$$

The top view of the field of view of the spatially distributed sensor array when M = 4 is shown in Figure 2b. It can be seen that the field of view of the spatially distributed sensor array is not a regular area. The solid angle formed by the intersection of the fields of view of Sensor 3 and Sensor 5 with those of Sensor 2 and Sensor 4 is defined as the field of view of the spatially distributed sensor array in this paper. Substituting γ=0 and Equation (15) into Formula (18), the field of view of the spatially distributed sensor array can be expressed as:(19)$$FOV=2\cos^{-1}\left(\frac{\cos\frac{\phi }{2}}{\cos\alpha }\right)$$

The relationship between the FOV of the spatially distributed sensor array and α of the spatially distributed sensor array is shown in the Figure 3. As α increases, the field of view of a spatially distributed sensor array decreases.

From Equation (8), we can see that the sensor combination matrix involved in the AOA solution needs to satisfy r(B)≥3. The spatially distributed sensor is a symmetrical structure. Therefore, Sensors 1, 2, and 3 are selected to analyze the impact of α on sensor measurement resolution. The azimuth and elevation angles can be expressed as:(20)$$\gamma =\tan^{-1}\left(\frac{P_{3}-P_{1}\cos\alpha }{\sqrt{{\left(P_{1}\sin\alpha \right)}^{2}+{\left(P_{2}-P_{1}\cos\alpha \right)}^{2}}}\right)=f_{\gamma }\left(P_{1},P_{2},P_{3}\right)$$(21)$$\beta =\tan^{-1}\left(\frac{P_{2}-P_{1}\cos\alpha }{P_{1}\sin\alpha }\right)=f_{\beta }\left(P_{1},P_{2}\right)$$

The resolution of a spatially distributed sensor array can be expressed as:(22)$$∆\gamma =\frac{\partial f_{\gamma }}{\partial P_{1}}∆P_{1}+\frac{\partial f_{\gamma }}{\partial P_{2}}∆P_{2}+\frac{\partial f_{\gamma }}{\partial P_{3}}∆P_{3}$$(23)$$∆\beta =\frac{\partial f_{\beta }}{\partial P_{1}}∆P_{1}+\frac{\partial f_{\beta }}{\partial P_{2}}∆P_{2}$$
where $∆P_{1}$, $∆P_{2}$, and $∆P_{3}$ represent the changes in sensor output power caused by changes in the incident angle. The resolution of the spatially distributed sensor array is determined by $∆P_{1}$, $∆P_{2}$, and $∆P_{3}$. However, according to Equation (2), the sensor output is proportional to the cosine of the incident angle. When β=0 and γ=0, small changes in the azimuth angle result in negligible changes in the output power of Sensors 1 and 3. Similarly, small changes in the elevation angle result in negligible changes in the output power of Sensors 1 and 2. In this case, the resolution of the spatially distributed sensor array is determined by a single sensor, leading to a relatively poor resolution, which can be expressed as:(24)$$∆\beta \approx \frac{\partial f_{\beta }}{\partial P_{2}}=\frac{∆P_{2}}{S}\frac{P_{1}\sin\alpha }{{\left(P_{1}\sin\alpha \right)}^{2}+{\left(P_{2}-P_{1}\cos\alpha \right)}^{2}}$$(25)$$∆\gamma \approx \frac{\partial f_{\gamma }}{\partial P_{3}}=\frac{∆P_{3}}{S}\frac{\sqrt{{\left(P_{1}\sin\alpha \right)}^{2}+{\left(P_{2}-P_{1}\cos\alpha \right)}^{2}}}{{\left(P_{1}\sin\alpha \right)}^{2}+{\left(P_{2}-P_{1}\cos\alpha \right)}^{2}+{\left(P_{3}-P_{1}\cos\alpha \right)}^{2}}$$

Substituting Equations (13)–(15) into Equations (24) and (25), the elevation and azimuth resolutions can be expressed as:(26)$$∆\beta \approx \frac{\cos\beta }{\sin\alpha \cos\gamma }\frac{∆P_{2}}{S}$$(27)$$∆\gamma \approx \frac{\cos\gamma }{\sin\alpha }\frac{∆P_{3}}{S}$$

$∆P_{2}$ and $∆P_{3}$ represent the minimum resolvable power provided by the sensors, and they are equal. When β=0 and γ=0, based on Equations (26) and (27), both the elevation and azimuth angles exhibit the same resolution. The relationship between angular resolution and α is shown in Figure 4. As α increases, the resolution of the spatially distributed sensor array improves.

Based on the above analysis, as α increases, the field of view decreases while the resolution improves. Conversely, as α decreases, the field of view increases and the resolution decreases. Considering both the field of view and angular resolution, this paper selects α=30° to achieve a field of view of 110° with a detection accuracy better than 1°. The spatially distributed sensor array designed in this paper is shown in Figure 5. The five sensors are placed on a 3D-printed panel with a size of 150 × 150 mm, and the total weight is only 140 g. The 3D printed panel consists of a top panel and four side panels, with α being 30°. After actual measurements, the error in α is within 0.1°.

### 2.3. Multi-Sensor Data Fusion Algorithm for the Spatially Distributed Sensor Array

Fluctuations in power density during beam propagation and interference from background light can introduce noise, leading to inaccuracies in the sensor output. When angle calculations are performed using only three sensors, significant angular errors may occur. In fact, the spatially distributed sensor array has sensor redundancy. As shown in Figure 6a, within field of view A and field of view B, data from multiple sensors can be utilized to conduct linear estimation of the AOA, thereby reducing errors caused by noise and improving the accuracy of the sensor output. Therefore, we propose a multi-sensor data fusion algorithm. Among the outputs of five sensors, the three maximum values are selected to calculate the AOA and estimate the direction of the incident beam. Different solution strategies are applied depending on the field of view in which the AOA is located. The flowchart of the multi-sensor data fusion algorithm is shown in Figure 6b.

Initially, the three maximum values in $\left\{P_{1}\right.\ldots P_{5}\}$ are substituted into Equations (8) and (9) to estimate AOA.Then, the following algorithm is employed based on the range of the estimated AOA.

I: If estimating the AOA within field of view A, any four power values can be selected from the five power values derived from the outputs of the five sensors, yielding five combinations. Each combination is substituted into Equations (8) and (9) to obtain five AOA solutions. By substituting all five power values into Equations (8) and (9), one AOA solution is obtained. The azimuth and elevation angles of the six solutions are then sorted in ascending order, with the maximum and minimum values removed, and the trimmed mean is calculated as β and γ.

II: If the estimated AOA is in the field of view B, the four maximum values in $\left\{P_{1}\right.\ldots P_{5}\}$ are substituted into Equations (8) and (9) to calculate β and γ.

III: If the estimated AOA is in the field of view C, the three maximum values in $\left\{P_{1}\right.\ldots P_{5}\}$ are substituted into Equations (8) and (9) to calculate β and γ.

## 3. Digital Processing of Spatially Distributed Sensor Array

The signal acquisition and data processing module of the AOA detection system using a spatially distributed sensor array are shown in Figure 7. A spatially distributed sensor array outputs five currents independently, which are converted into voltage signals. After amplification, they are sent to the A/D converter (ADC) and converted into digital signals. The microcontroller reads values from five channels and transmits them to the computer via a USB port, where they are processed using a multi-sensor data fusion algorithm to calculate the AOA. In future research, we plan to use FPGA boards for AOA calculation to improve the output bandwidth of the spatially distributed sensor array.

## 4. Experiment Results

In order to analyze the angular measurement performance of the developed spatially distributed sensor array, we constructed an indoor angle of arrival testing platform. The experimental system uses a 1550 nm fiber laser as the light source, as shown in Figure 8a. The divergence angle of the beam is 5°, and the distance to the spatially distributed sensor array is 6.5 m, with the radius of the light spot reaching the sensor array being approximately 570 mm. The ADC output data width is 16 bits. The spatially distributed sensor array is mounted on a two-dimensional gimbal with an accuracy of 0.001°. By adjusting the angles of the gimbal, the orientation of the sensor can be altered, enabling the measurement of the incident angle of the beam, as shown in Figure 8b.

Before the testing, the primary sources of errors in the angle of arrival detection system of the spatially distributed sensor array were analyzed, and the error calibration was implemented. Firstly, this work uses five identical photodetectors and photoelectric conversion circuit modules. However, sensor response non-uniformity and differences in circuit signal gains can affect the sensor outputs, thus reducing the measurement accuracy of the AOA. Secondly, the sensor installation error arising from the fixation of the photoelectric conversion circuit module on the 3D-printed panel, along with the error in α, can result in sensor position error.

The spatially distributed sensor array is installed in a darkroom without light. The output voltage of each sensor is generated by dark current. The dark current noise in each sensor can be considered uniform, so the difference in circuit signal gain of each sensor circuit can be calibrated to obtain the gain correction coefficient of each sensor. The laser light source is turned on, and five sensors are placed one after another at the position of panel 1 in Figure 8b. Each sensor is irradiated by a light source with the same energy. By varying the intensity of the incident light, the actual optical response of each sensor is calibrated based on the changes in the output voltage.

Taking the position of sensor 1 as a reference, by comparing the angle difference between the actual output and theoretical output of each sensor at different incident angles, the position errors of the five sensors in the azimuth and elevation directions are analyzed. The normal vector coordinate matrix of the spatially distributed sensor array after correction is expressed as:(28)$$K^{T}=\left[\begin{array}{c} n_{1}\\ n_{2}\\ \begin{array}{c} n_{3}\\ \begin{array}{c} n_{4}\\ n_{5} \end{array} \end{array} \end{array}\right]=\left[\begin{array}{c} 1\\ \cos\left(\alpha {+\sigma }_{y2}\right)\cos\sigma_{z2}\\ \cos\sigma_{y3}\cos\left(\alpha +\sigma_{z3}\right)\\ \cos\left(\alpha {+\sigma }_{y4}\right)\cos\sigma_{z4}\\ \cos\sigma_{y5}\cos\left(\alpha +\sigma_{z5}\right) \end{array}\begin{array}{c} 0\\ \sin\left(\alpha {+\sigma }_{y2}\right)\cos\sigma_{z2}\\ \sin\sigma_{y3}\cos\left(\alpha +\sigma_{z3}\right)\\ -\sin\left(\alpha {+\sigma }_{y4}\right)\cos\sigma_{z4}\\ \sin\sigma_{y5}\cos\left(\alpha +\sigma_{z5}\right) \end{array}\begin{array}{c} 0\\ \sin\sigma_{z2}\\ \sin(\alpha +\sigma_{z3})\\ \sin\sigma_{z4}\\ -\sin(\alpha +\sigma_{z5}) \end{array}\right]$$
where $\sigma_{y}$ is the angle error in the azimuth direction for each sensor, and $\sigma_{z}$ is the angle error in the elevation direction.

Due to the long propagation distance of light in free space and the beam diameter being much larger than the diameter of the spatial distribution sensor, the AOA detection algorithm for the spatial distribution sensor array is based on the assumption of a uniform energy distribution of the light spot. However, in this experiment, where the distance between the light source and the spatially distributed sensor array is relatively short, it is necessary to correct the sensor output based on the intensity distribution of the Gaussian beam.

The normalized output of each sensor is shown in Figure 9a, with the elevation angle set to 0° and the azimuth angle moving from −60° to 60° in 1° intervals, and Figure 9b demonstrates the normalized output of five sensors after calibration, which significantly improves the measurement accuracy of the sensors.

After correcting the systematic errors of the spatially distributed sensor array, the multi-sensor data fusion algorithm was used to calculate the azimuth and elevation angles within a 110° field of view. Figure 10 shows the calculated AOA results of five measurements at 5° intervals of azimuth angle from −55° to 55° at elevation angles of 0°, 25°, and 50°. The results show that they are close to the actual values, with mean angle errors of 0.44°, 0.58°, and 2.15° for azimuth and 0.4°, 0.85°, and 2.14° for elevation. In the fields of view A and B, higher measurement accuracy is achieved by fusing the outputs of multiple sensors. Repeatability results with the elevation angle set to 0° and the azimuth angle moving from −50° to 50° in 5° intervals are shown in Figure 11, which is better than 0.15°.

The 2D AOA detection performance of the system is measured, simultaneously verifying the effectiveness of the multi-sensor data fusion algorithm. The test results are shown in Figure 12. After using the data fusion algorithm, the maximum errors of azimuth and elevation angle in field of view A are 1.59° and 1.7°, respectively, and the RMSE is 0.6° and 0.67°. The maximum measurements of azimuth and elevation angle in the 110° field of view are 4.3° and 6.7°, respectively, with RMSE of 1.1° and 1.7°. In the fields of view A and B, higher measurement accuracy is achieved by fusing the outputs of multiple sensors. The 2D experimental results demonstrate that the spatially distributed sensor array exhibits high accuracy within a large field of view.

However, conventional spot position detection systems often require complex optical components, such as fisheye lenses, high-resolution CCDs, and multi-unit sensor arrays, to achieve a wide field of view [9,10]. The fisheye lens, composed of multiple optical elements, contributes significantly to the system’s overall volume and weight. Furthermore, the associated processing circuitry tends to be more complex, resulting in higher power consumption. In contrast, the spatially distributed sensor array proposed in this study positions the sensors on a 150 mm × 150 mm panel, with a total mass of merely 140 g. This design not only simplifies the circuitry but also substantially reduces power consumption, presenting a more compact, lightweight, and energy-efficient solution.

## 5. Conclusions

This paper develops an AOA detection model for the spatially distributed sensor array and analyzes the impact of structural design on the field of view and detection accuracy. Based on it, a spatially distributed sensor array was designed, consisting of five InGaAs PIN photodiodes mounted on a 3D-printed panel with a 30° angle between the top panel and the side panels. A multi-sensor data fusion algorithm is proposed to mitigate measurement errors caused by the random variations in power density during the beam propagation process. The accuracy and repeatability of two-dimensional AOA detection were evaluated by indoor experiments. The results demonstrate that within a 60° field of view, the measurement accuracy of AOA is significantly improved, with a maximum angular error of 1.7° and an RMSE of 0.67°, validating the effectiveness of the multi-sensor data fusion algorithm. Compared to the systems that detect the AOA based on light spot position, the proposed design significantly reduces the SWaP, which has broad application prospects in future multi-satellite laser communication.

Furthermore, we will conduct multi-environment optimization design and testing. By encoding the received beam and designing filters to improve the signal-to-noise ratio (SNR), the performance of the sensor will be further enhanced.

## Figures and Tables

**Figure 1 sensors-25-01625-f001:**
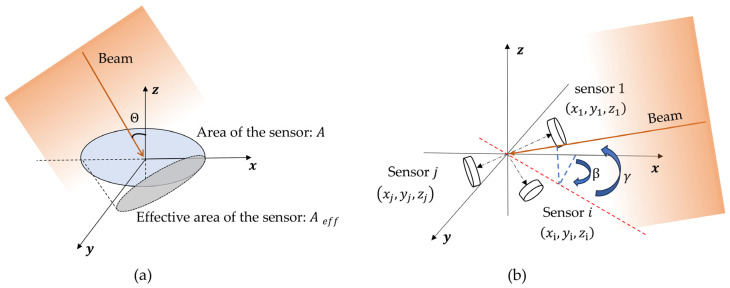
(**a**) Schematic diagram of the relationship between sensor effective area and incident light. (**b**) Schematic diagram of sensor position and AOA.

**Figure 2 sensors-25-01625-f002:**
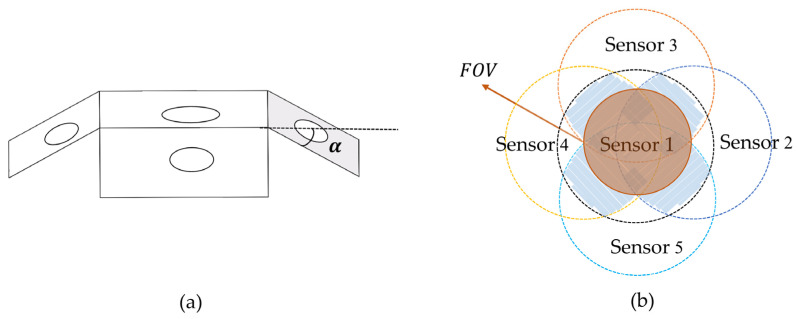
(**a**) Schematic diagram of a spatially distributed sensor array when M=4. (**b**) Field of view of spatially distributed sensor array when M=4.

**Figure 3 sensors-25-01625-f003:**
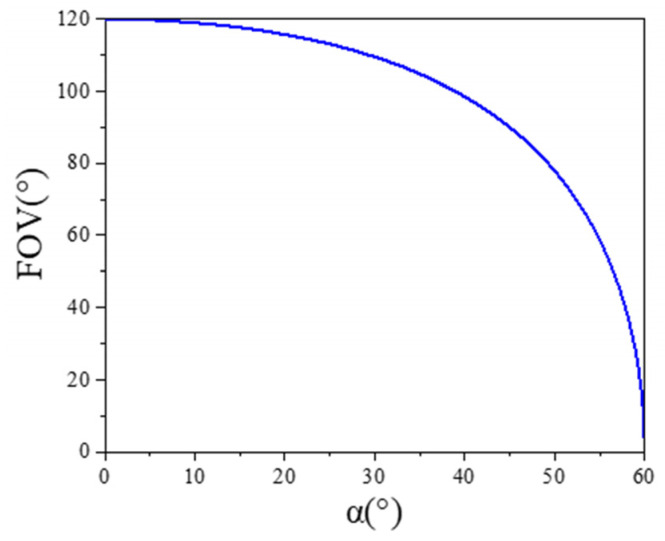
Schematic diagram of the relationship between the FOV of the spatially distributed sensor array and α.

**Figure 4 sensors-25-01625-f004:**
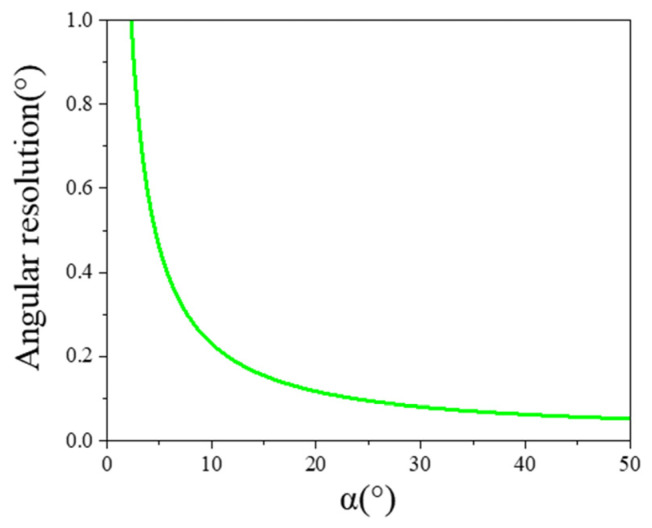
Schematic diagram of the relationship between angular resolution of the spatial distribution sensor and α.

**Figure 5 sensors-25-01625-f005:**
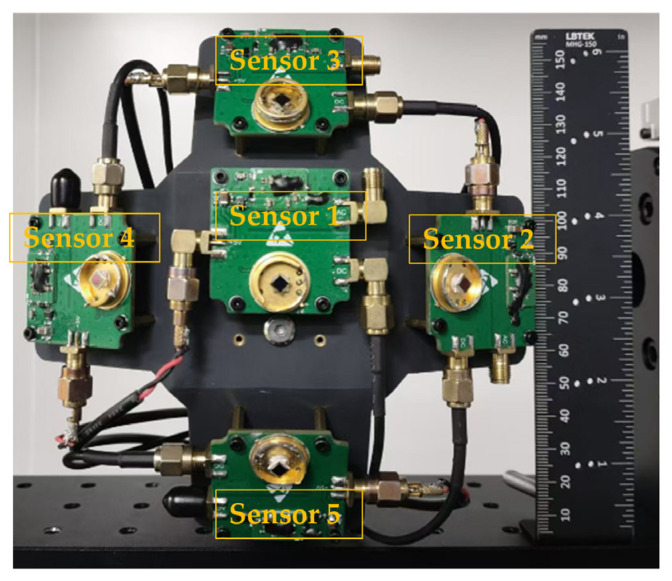
Image of the spatially distributed sensor array.

**Figure 6 sensors-25-01625-f006:**
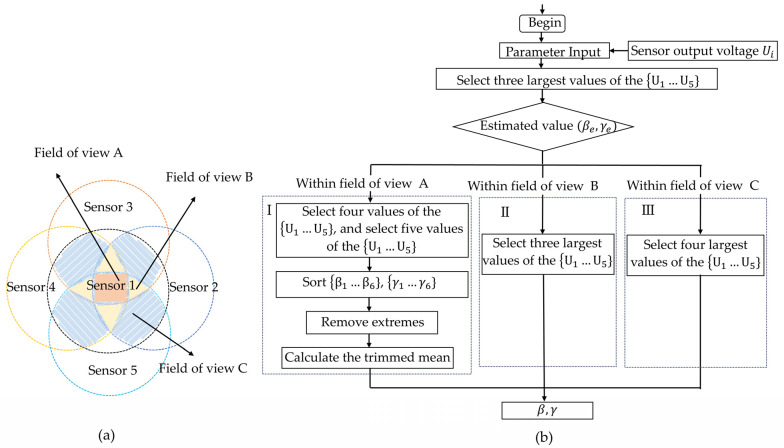
(**a**) Schematic diagram of the field of view division of a spatially distributed sensor array. (**b**) Multi-sensor data fusion algorithm flow chart.

**Figure 7 sensors-25-01625-f007:**
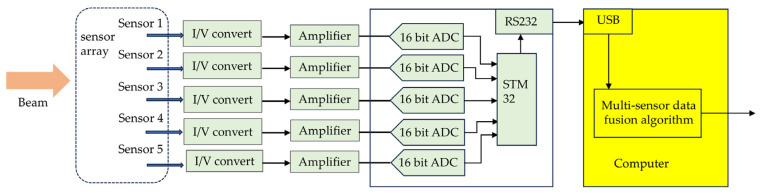
Schematic circuit diagram of digital processing.

**Figure 8 sensors-25-01625-f008:**
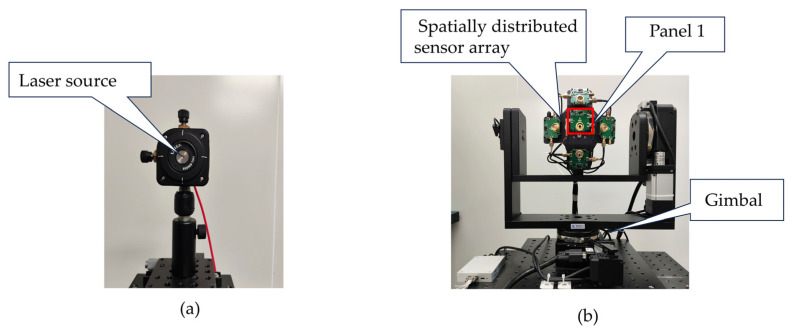
The experimental platform. (**a**) Laser source. (**b**) Spatially distributed sensor array and two-dimensional gimbal.

**Figure 9 sensors-25-01625-f009:**
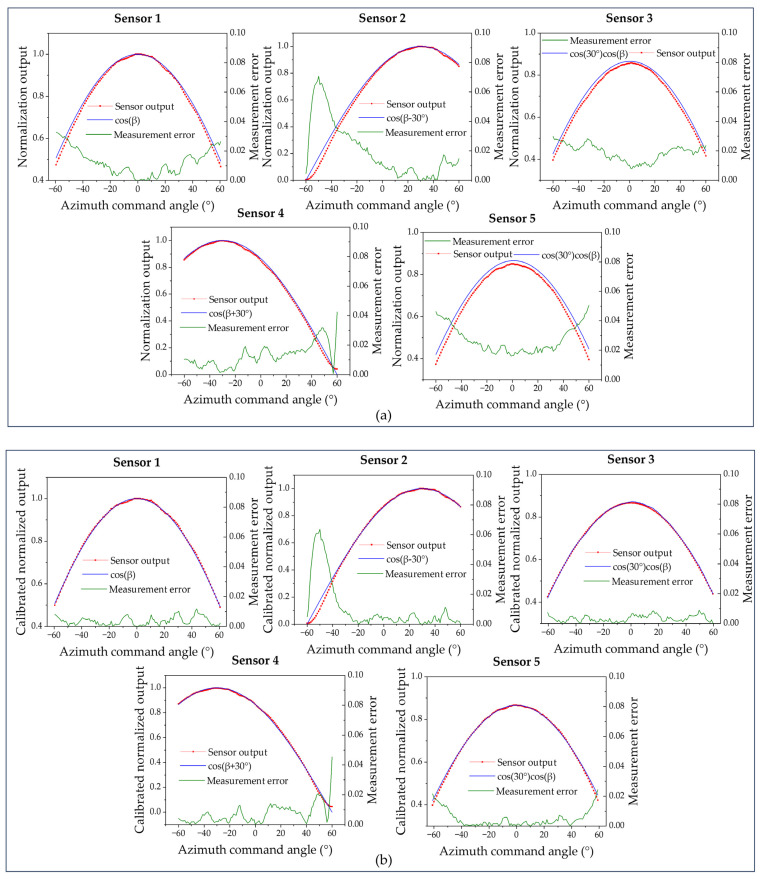
(**a**) Normalized output of five sensors. (**b**) Normalized output of five sensors after error calibration.

**Figure 10 sensors-25-01625-f010:**
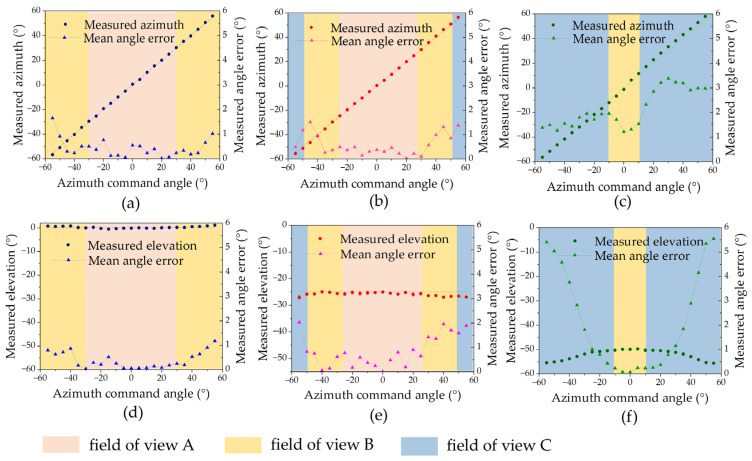
AOA detection results at different elevation angles after error correction. (**a**) The calculated azimuth coordinate curves when γ=0°. (**b**) The calculated azimuth coordinate curves when γ=25°. (**c**) The calculated azimuth coordinate curves when γ=50°. (**d**) The calculated elevation coordinate curves when γ=0°. (**e**) The calculated elevation coordinate curves when γ=25°. (**f**) The calculated elevation coordinate curves when γ=50°.

**Figure 11 sensors-25-01625-f011:**
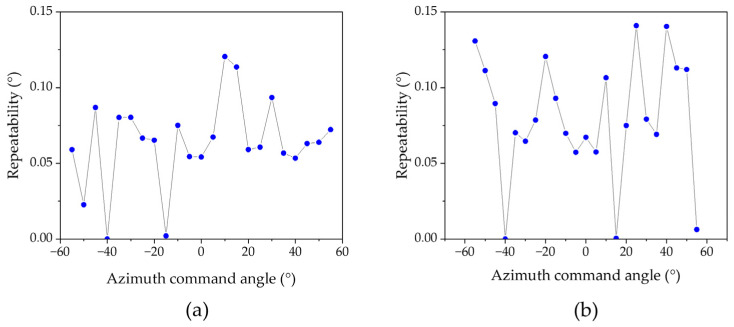
Repeatability results with the elevation angle set to 0° and the azimuth angle moving from −50° to 50° in 5° intervals. (**a**) Repeatability results for azimuth. (**b**) Repeatability results for elevation.

**Figure 12 sensors-25-01625-f012:**
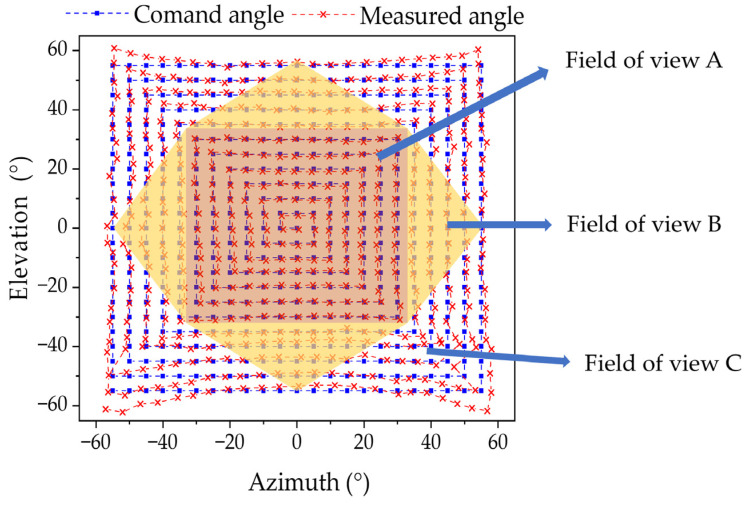
Two-dimensional experimental AOA results with the spatially distributed sensor array.

## Data Availability

Data are contained within the article.

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
