# Peer review of "Angle of Arrival for the Beam Detection Method of Spatially Distributed Sensor Array"

_sensors, 2025, doi:10.3390/s25051625_

Round 1
Reviewer 1 Report
Comments and Suggestions for Authors
In this manuscript, authors demonstrate a system which uses a spatially distributed sensor array to detect the AOA in a wide FOV. The designed spatially distributed sensor array has the advantages of a wide FOV and low size, weight, and power (SWaP), presenting great potential for multi-satellite laser communication applications. This work can be considered for publication after the following issues.
- Does the experimental validation cover system performance under various environmental conditions?
- What are the primary sources of measurement error, and how can these errors be further reduced?
- What are the specific advantages of this design compared to other existing AOA detection methods?
- How do data fusion algorithms improve the measurement accuracy of AOA detection systems?
Author Response
Dear Reviewer,
We would like to sincerely thank you for taking the time to review our manuscript and provide valuable feedback. We greatly appreciate your constructive comments and suggestions, which have significantly helped improve the quality of our work. We provide a detailed response to your comments and outline the revisions made to the manuscript.
Comments 1: Does the experimental validation cover system performance under various environmental conditions?
Response 1: We sincerely appreciate the reviewer for pointing this out. This paper presents the results of indoor experiments. However, in the future, we will conduct multi-environment optimization design and testing. By encoding the received beam and designing filters to improve the signal-to-noise ratio (SNR), the performance of the sensor will be further enhanced. We have added this in lines 326-328 of the manuscript.
Comments 2: What are the primary sources of measurement error, and how can these errors be further reduced?
Response 2: Thank you for this valuable feedback. The primary sources of measurement errors include sensor response uniformity, differences in circuit signal gains and sensor position errors. By calibrating the errors, the angular measurement accuracy of the AOA can be improved. We have rewritten the main error sources in lines 234-241. The implementation of errors calibration is described in lines 242-256.
Comments 3: What are the specific advantages of this design compared to other existing AOA detection methods?
Response 3: We have carefully considered this comment and have made the following improvements. Compared to the AOA detection system based on spot position, the spatially distributed sensor array can achieve AOA detection in a wide field of view without a fisheye lens or larger array scales. At the same time, it reduces the system's size, weight, and power consumption. We have described the advantages compared to the spot position detection system in lines 55-57 of the paper. Compared to other studies on spatially distributed sensor arrays, the main advantages of this design are analyzing the impact of key parameters of the spatially distributed sensor on the field of view and angular resolution, and designing the spatially distributed sensor array based on this analysis. A multi-sensor data fusion algorithm is proposed to enhance measurement accuracy, and the sources of errors are analyzed and corrected. We have rewritten lines 68–80 and highlighted the differences between previous studies and our design approach.
Comments 4: How do data fusion algorithms improve the measurement accuracy of AOA detection systems?
Response 4: Thank you for your thoughtful question. The multi-sensor data fusion algorithm improves measurement accuracy by mitigating the effect of noise. Fluctuations in power density during beam propagation and interference from background light can introduce noise, leading to inaccuracies in the sensor output. When angle calculations are performed using only three sensors, significant angular errors may occur. In fact, the spatially distributed sensor array has sensor redundancy. within field of view A and field of view B, data from multiple sensors can be utilized to conduct linear estimation of the AOA, thereby reducing errors caused by noise and improving the accuracy of the sensor output. Therefore, we propose a multi-sensor data fusion algorithm. We have revised the description in lines 182-191.

Reviewer 2 Report
Comments and Suggestions for Authors
The authors present a method for detecting the Angle of Arrival (AOA) using a spatially distributed sensor array for wide FOV application in optical satellite communication. The formulation and results are well presented. However, the reviewer has a few concerns as follows:
- The formulation is well structured, but it is based on existing known AOA detection principles and equations. In order to emphasize the contribution of the work and highlight its significance, the authors are encouraged to make it further clarify which aspects of the formulation are existing detection principles (with proper citation) and which constitute their contribution.
- The authors claim to provide a systematic design approach. However, the authors are recommended to clarify further what aspects their proposed design approach differentiates from existing ones, mainly previous work cited in the paper.
- The authors state in the conclusion that "Compared to existing systems that detect the AOA based on beam spot position, the proposed design significantly reduces SWaP, which has broad application prospects in future multi-satellite laser communication." The authors are encouraged to support this claim with reported metrics (e.g., FOV, accuracy, SWaP) in cited previous work.
- In the introduction, the authors state that "Finally, calibration methods are introduced to correct sensor response uniformity and installation position errors, and the effectiveness...." However, the paper does not introduce calibration methods but rather discusses their implementation. The authors are encouraged to rephrase this claim to accurately reflect that.
- In line 235, the authors state: "....fixed onto the 3D-printed panel using screws, which may cause mounting errors. This paper calibrates both of the system errors to improve measurement accuracy." It is not clear what is meant by "this paper calibrates".
Author Response
Dear Reviewer,
We would like to sincerely thank you for taking the time to review our manuscript and provide valuable feedback. We greatly appreciate your constructive comments and suggestions, which have significantly helped improve the quality of our work. We provide a detailed response to your comments and outline the revisions made to the manuscript.
Comments 1: The formulation is well structured, but it is based on existing known AOA detection principles and equations. In order to emphasize the contribution of the work and highlight its significance, the authors are encouraged to make it further clarify which aspects of the formulation are existing detection principles (with proper citation) and which constitute their contribution.
Response 1: Thank you for pointing this out. The basic measurement principle of spatial distribution sensor is in line 85, citing the description in reference 14 that "the output of the sensor is related to the effective receiving area of the sensor". The arrival angle calculation formula in line 109 is quoted from reference 18.
Comments 2: The authors claim to provide a systematic design approach. However, the authors are recommended to clarify further what aspects their proposed design approach differentiates from existing ones, mainly previous work cited in the paper.
Response 2: Thank you for suggesting this improvement. Previous studies have clarified the AOA detection principle of spatially distributed sensor arrays, developed prototypes, and conducted verification experiments, but the impact and optimization design of key parameters of such arrays, data processing methods, and error analysis are not in-depth enough. There is also a lack of systematic design methods for spatially distributed sensor arrays. The main contributions of this paper are analyzing the impact of key parameters of the spatially distributed sensor on the field of view and angular resolution, and designing the spatially distributed sensor array based on this analysis. A multi-sensor data fusion algorithm is proposed to enhance measurement accuracy. Finally, the main sources of measurement errors in the spatially distributed sensor array were discussed, and error calibration was implemented. We have rewritten lines 68–80 and highlighted the differences between previous studies and our design approach.
Comments 3: The authors state in the conclusion that "Compared to existing systems that detect the AOA based on beam spot position, the proposed design significantly reduces SWaP, which has broad application prospects in future multi-satellite laser communication." The authors are encouraged to support this claim with reported metrics (e.g., FOV, accuracy, SWaP) in cited previous work.
Response 3: Thank you for this valuable feedback. We have added a comparison with a light spot position detection system to support our conclusion: “Conventional spot position detection systems often require complex optical components, such as fisheye lenses, high-resolution CCDs, and multi-unit sensor arrays, to achieve a wide field of view [9,10]. The fisheye lens, composed of multiple optical elements, composed of multiple optical elements, contribute significantly to the system's overall volume and weight. Furthermore, the associated processing circuitry tends to be more complex, resulting in higher power consumption. In contrast, the spatially distributed sensor array proposed in this study positions the sensors on a 150 mm × 150 mm panel, with a total mass of merely 140 g. This design not only simplifies the circuitry but also substantially reduces power consumption, presenting a more compact, lightweight, and energy-efficient solution.” This part of the content is in lines 302-311 of the paper.
Comments 4: In the introduction, the authors state that "Finally, calibration methods are introduced to correct sensor response uniformity and installation position errors, and the effectiveness...." However, the paper does not introduce calibration methods but rather discusses their implementation. The authors are encouraged to rephrase this claim to accurately reflect that.
Response 4: We agree with the reviewer’s comment. We have revised the sentence “Finally, calibration methods are introduced to correct sensor response uniformity and installation position errors, and the effectiveness …” to “Finally, the primary sources of measurement errors in the spatially distributed sensor array were discussed, and error calibration was implemented.” We have rewritten this part in lines 77-79.
Comments 5: In line 235, the authors state: "....fixed onto the 3D-printed panel using screws, which may cause mounting errors. This paper calibrates both of the system errors to improve measurement accuracy." It is not clear what is meant by "this paper calibrates".
Response 5: We apologize for not describing it clearly. Here, we aim to explain the main error sources of the AOA detection system of the spatially distributed sensor array, and the error calibration was implemented. We have rewritten in lines 234-236.

Reviewer 3 Report
Comments and Suggestions for Authors
In this manuscript, laser space network is an important development direction for intersatellite communication. Detecting the angle of arrival (AOA) of multiple satellites in a wide field of view (FOV) is the key to realizing inter-satellite laser communication networking. The traditional AOA detection method based on the lens system has a limited FOV. In this paper, we demonstrate a system that uses a spatially distributed sensor array to detect the AOA in a wide FOV. The basic concept is to detect AOA using the signal strength of each sensor at different spatial angles. An AOA detection model was developed, and the relationship of key structural parameters of the spatially distributed sensor array on the FOV and detection accuracy was analyzed. Furthermore, a spatially distributed sensor array prototype consisting of 5 InGaAs PIN photodiodes distributed on a 3D-printed structure with an inclination angle of 30° was developed. In order to improve the angle calculation accuracy, a multisensor data fusion algorithm is proposed. The experimental results show that the prototype’s maximum FOV is 110°. The root mean square error (RMSE) for azimuth is 0.6° within a 60° FOV, whereas the RMSE for elevation is 0.67°. The RMSE increases to 1.1° for azimuth and 1.7° for elevation when the FOV expands to 110°. The designed spatially distributed sensor array has the advantages of a wide FOV and low size, weight, and power (SWaP), presenting great potential for multi-satellite laser communication applications.
This topic is interesting, I thinks this manuscript can be accepted for publication after minor revision.
- In the line 20 of page 1, “a inclination angle” should be “an inclination angle”.
- Ai, cos and sin in all equations should be regular script.
- The blank space before “where” after Eqs, (1), (2), (3)(5),(6) , (28) should be deleted, “Where” should be “where”,
- All the figures should be clear.
In this manuscript, laser space network is an important development direction for intersatellite communication. Detecting the angle of arrival (AOA) of multiple satellites in a wide field of view (FOV) is the key to realizing inter-satellite laser communication networking. The traditional AOA detection method based on the lens system has a limited FOV. In this paper, we demonstrate a system that uses a spatially distributed sensor array to detect the AOA in a wide FOV. The basic concept is to detect AOA using the signal strength of each sensor at different spatial angles. An AOA detection model was developed, and the relationship of key structural parameters of the spatially distributed sensor array on the FOV and detection accuracy was analyzed. Furthermore, a spatially distributed sensor array prototype consisting of 5 InGaAs PIN photodiodes distributed on a 3D-printed structure with an inclination angle of 30° was developed. In order to improve the angle calculation accuracy, a multisensor data fusion algorithm is proposed. The experimental results show that the prototype’s maximum FOV is 110°. The root mean square error (RMSE) for azimuth is 0.6° within a 60° FOV, whereas the RMSE for elevation is 0.67°. The RMSE increases to 1.1° for azimuth and 1.7° for elevation when the FOV expands to 110°. The designed spatially distributed sensor array has the advantages of a wide FOV and low size, weight, and power (SWaP), presenting great potential for multi-satellite laser communication applications.
This topic is interesting, I thinks this manuscript can be accepted for publication after minor revision.
- In the line 20 of page 1, “a inclination angle” should be “an inclination angle”.
- Ai, cos and sin in all equations should be regular script.
- The blank space before “where” after Eqs, (1), (2), (3)(5),(6) , (28) should be deleted, “Where” should be “where”,
- All the figures should be clear.
Author Response
Dear Reviewer
We would like to sincerely thank you for taking the time to review our manuscript and provide valuable feedback. We greatly appreciate your constructive comments and suggestions, which have significantly helped improve the quality of our work. We provide a detailed response to your comments and outline the revisions made to the manuscript.
Comments 1: In the line 20 of page 1, “a inclination angle” should be “an inclination angle”.
Response 1: We appreciate the reviewer pointing this out. In the line 20 of page 1, “a inclination angle” have corrected “an inclination angle”.
Comments 2: Ai, cos and sin in all equations should be regular script.
Response 2: We appreciate this valuable feedback. Cos and sin in all equations are corrected in regular script.
Comments 3: The blank space before “where” after Eqs, (1), (2), (3),(5), (6), (28) should be deleted, “Where” should be “where”,
Response 3: Thank you for pointing this out. We have deleted the blank space before “where” after Eqs, (1), (2), (3), (5), (6), (28),and “Where” have corrected “where”.
Comments 4: All the figures should be clear.
Response 4: Thank you for this suggestion. We have adjusted Figures 1, 2, 6, 9, and 10 to make them clearer.

Reviewer 4 Report
Comments and Suggestions for Authors
The paper aims at developing of the scheme for angle of arriving detection by mean the spatially distributed array of light sensors. The idea of angle of arriving measurements by spatially distributed sensors is not new, which is indicated in the introduction. For example, similar geometry was described in Ref [14] of the current paper. However, the angel estimation errors in [14] appeared to be several times bigger than in the current manuscript. The main advantage of the current paper is a rather deep analysis of the problem comparing with similar works and more precise results with rather small errors. I suggest here to emphasize the difference between this paper and other available works on AOA measurements by means the spatially distributed sensors.
The sensor in this paper is 15 times bigger than sensor described in the [14]. Does the difference in sizes play a role? I am also interested in measured laser beam parameters such as divergence and spot size at the detection area, which are not indicated in the manuscript and can be important during such a system utilizing.
Another question about light to be detected: can it be modulated to bring some additional information, e.g., about ID of the emitting device?
Measurements in this paper were performed in the dark room. How can the light of the Sun affect this system performance?
Nothing is said about accuracy of sensor positioning angle alpha. 3D printing accuracy is far from ideal. May be something like this was mentioned after Fig. 8, however, it is not clear.
The paragraph after Figure 8 seems to be not connected with surrounding text. It is better to rewrite it.
The work is not free of shortcomings. The English must be checked more carefully. The text is full of grammar mistakes; there are some confusing phrases and misprints.
Abbreviations indicated in the Abstract better to be introduced again in the text (AOA, FOV, SWaP, etc.).
Line 13. “key to realizing”->”key for realizing” or “key to realize”
Line 37. “to fulfill this problem” -> “to address this problem”
Fig. 6. The flow chart is better to be enlarged.
Fig. 8. “leaser source” -> “laser source”
Lines 240-241. The sentence “Turn on …. Figure 8b” should be rewritten.
Figure 9 is too small.
Figure 9 and 10. Does the “Command Angle” mean the angle Betta? If so it is better to indicated in the each plot label.
I recommend the paper for publication after minor revision and answering my questions.
The English must be checked more carefully. The text is full of grammar mistakes; there are some confusing phrases and misprints.
Author Response
Dear Reviewer
We would like to sincerely thank you for taking the time to review our manuscript and provide valuable feedback. We greatly appreciate your constructive comments and suggestions, which have significantly helped improve the quality of our work. We provide a detailed response to your comments and outline the revisions made to the manuscript.
Comments 1: The paper aims at developing of the scheme for angle of arriving detection by mean the spatially distributed array of light sensors. The idea of angle of arriving measurements by spatially distributed sensors is not new, which is indicated in the introduction. For example, similar geometry was described in Ref [14] of the current paper. However, the angel estimation errors in [14] appeared to be several times bigger than in the current manuscript. The main advantage of the current paper is a rather deep analysis of the problem comparing with similar works and more precise results with rather small errors. I suggest here to emphasize the difference between this paper and other available works on AOA measurements by means the spatially distributed sensors.
Response 1: Thank you for suggesting this improvement. Previous studies have clarified the AOA detection principle of spatially distributed sensor arrays, developed prototypes, and conducted verification experiments, but the impact and optimization design of key parameters of such arrays, data processing methods, and error analysis are not in-depth enough. There is also a lack of systematic design methods for spatially distributed sensor arrays. The main contributions of this paper are analyzing the impact of key parameters of the spatially distributed sensor on the field of view and angular resolution, and designing the spatially distributed sensor array based on this analysis. A multi-sensor data fusion algorithm is proposed to enhance measurement accuracy. Finally, the main sources of measurement errors in the spatially distributed sensor array were discussed, and error calibration was implemented. We have rewritten lines 68–80 and highlighted the differences between previous studies and our design approach.
Comments 2: The sensor in this paper is 15 times bigger than sensor described in the [14]. Does the difference in sizes play a role? I am also interested in measured laser beam parameters such as divergence and spot size at the detection area, which are not indicated in the manuscript and can be important during such a system utilizing.
Response 2: We appreciate the reviewer pointing this out. From the perspective of detection principles, the size does not affect the angle of arrival detection of the spatially distributed sensor array. However, when the light spot size is smaller than the sensor array's size, AOA detection is not possible. The application scenario in this paper is low-earth orbit satellite communication, where the beam spot becomes much larger than the detector size after long-distance transmission. This is a prototype, and in future work, we will optimize the size of the spatially distributed sensor array in coordination with other modules. We add a description of the laser light source: "The divergence angle of the beam is 5°, and the distance to the spatially distributed sensor array is 6.5 m, with the radius of the light spot reaching the sensor array being approximately 570 mm." This part of the content is in lines 223-227 of the paper.
Comments 3: Another question about light to be detected: can it be modulated to bring some additional information, e.g., about ID of the emitting device?
Response 3: We have carefully considered this question. The light beam can be modulated. The sensor response bandwidth selected in this paper is 3MHz, which can achieve low-rate communication. In future multi-satellite communications, the satellite number currently connected can be identified by modulating the optical signal. This is also our future work. In lines 326-328, we outline future work.
Comments 4: Measurements in this paper were performed in the dark room. How can the light of the Sun affect this system performance?
Response 4: We sincerely appreciate the reviewer for pointing this out. This paper presents the results of indoor experiments. However, in the future, we will conduct multi-environment optimization design and testing. By encoding the received beam and designing filters to improve the signal-to-noise ratio (SNR), the performance of the sensor will be further enhanced. We have added this in lines 326-328 of the manuscript.
Comments 5: Nothing is said about accuracy of sensor positioning angle alpha. 3D printing accuracy is far from ideal. May be something like this was mentioned after Fig. 8, however, it is not clear.
Response 5: We agree with the reviewer's observation and have added that the measured error of α is within 0.1° in line 178. In lines 239-241, the sensor position error is analyzed: “the sensor installation error arising from the fixation of the photoelectric conversion circuit module on the 3D-printed panel, along with the errors in α, can result in sensor position error.” and can be corrected by formula 28
Comments 6: The paragraph after Figure 8 seems to be not connected with surrounding text. It is better to rewrite it.
Response 6: Thank you for this suggestion. In the paragraph after Figure 8, we aim to explain the primary error sources of the AOA detection system of the spatially distributed sensor array, and the error calibration was implemented. We have rewritten the paragraph in lines 234-241 of the manuscript.
Comments 7: The work is not free of shortcomings. The English must be checked more carefully. The text is full of grammar mistakes; there are some confusing phrases and misprints. Abbreviations indicated in the Abstract better to be introduced again in the text (AOA, FOV, SWaP, etc.).
Response 7: We are very grateful to the reviewer for pointing out the problem. We have reintroduced the abbreviations that appeared in the abstract into the main text, corrected the grammatical errors in the main text, and modified some figures.
Line 13. “key to realizing” have corrected “key to realize”.
Line 37. “to fulfill this problem” have corrected “to address this problem”.
Fig. 6. The flow chart has enlarged.
Fig. 8. “leaser source” have corrected “laser source”
Lines 245-246. We have revised the sentence “Turn on …. Figure 8b” to “Turn on the laser light source, and five sensors are placed one after another at the position of panel 1 in Figure 8b.”
Figure 9 has enlarged.
Figure 9 and 10. “Command angle” have corrected “Azimuth command angle”.
